# Widespread reworking of Hadean-to-Eoarchean continents during Earth's thermal peak

C. L. Kirkland [1✉], M. I. H. Hartnady[1], M. Barham [1], H. K. H. Olierook [1], A. Steenfelt[2] & J. A. Hollis[3]

The nature and evolution of Earth's crust during the Hadean and Eoarchean is largely unknown owing to a paucity of material preserved from this period. However, clues may be found in the chemical composition of refractory minerals that initially grew in primordial material but were subsequently incorporated into younger rocks and sediment during lithospheric reworking. Here we report Hf isotopic data in 3.9 to 1.8 billion year old detrital zircon from modern stream sediment samples from West Greenland, which document successive reworking of felsic Hadean-to-Eoarchean crust during subsequent periods of magmatism. Combined with global zircon Hf data, we show a planetary shift towards, on average, more juvenile Hf values 3.2 to 3.0 billion years ago. This crustal rejuvenation was coincident with peak mantle potential temperatures that imply greater degrees of mantle melting and injection of hot mafic-ultramafic magmas into older Hadean-to-Eoarchean felsic crust at this time. Given the repeated recognition of felsic Hadean-to-Eoarchean diluted signatures, ancient crust appears to have acted as buoyant life-rafts with enhanced preservation-potential that facilitated later rapid crustal growth during the Meso-and-Neoarchean.

[1] Timescales of Mineral Systems Group, School of Earth and Planetary Sciences, Curtin University, Perth, WA 6102, Australia. [2] The Geological Survey of Denmark and Greenland, Øster Voldgade 10, 1350 Copenhagen K, Denmark. [3] Department of Geology, Ministry of Mineral Resources, Government of Greenland, P.O. Box 930, 3900 Nuuk, Greenland. ✉email: c.kirkland@curtin.edu.au

Despite decades of study, fundamental aspects of the development of Earth's oldest continental crust remain enigmatic. Various mechanisms have been proposed to explain Archean crust formation including via mantle plumes[1], density foundering and melt generation[2], or bolide impacts[3]. Models for crust production over time are also variable, depending on the balance between new crust extraction from the mantle versus recycling of old crust into the mantle, with the equilibrium between these processes either mediated or expressed by tectonics[4]. Crust production rates have thus been regarded as key in understanding secular change in tectonic setting[5]. For example, early continental crust before 3.0 billion years ago (Ga) may have formed via vertical processes[6] but may additionally have formed via horizontal processes akin to present-day plate tectonics[7]. Pre-existing continental crust may also play an important role in nucleating further growth[8,9]. Thus growth of continental crust is closely coupled with plate tectonics and magmatic fractionation, but the rate and mechanism of crust production toward the present continents remains uncertain.

Tracking the evolution of our oldest continental crust has commonly employed radiogenic isotopes such as Lu-Hf, which resolve magmatic source region characteristics of dated crystalline basement and can be recast to a measure of mantle extraction[10]. Such Hf isotope signatures, fundamentally reflecting a fractionation event between parent and daughter elements in the source magma, can be used to address terrane affinity and rates of crust production. Deeper into Earth's history, there is inevitably a progressively more fragmentary geological record as a consequence of loss of crystalline basement through crustal reworking and erosion, which serves to reduce the temporal fidelity of any record. However, sedimentary rocks preserve an important archive of crustal evolution by storing the denuded remains of crystalline source rocks that may no longer be exposed, or even extant. To track crustal evolution, detrital zircon Hf isotopes have proved to be particularly useful because of the mineral's physical and chemical robustness, while also recording time encoded source information[11,12].

In this work, we present a new zircon U-Pb and Hf isotope dataset from stream sediment from a Mesoarchean to Neoarchean part of the North Atlantic Craton (NAC), southwest Greenland that records hitherto unrecognized >3.2 Ga components (Fig. 1; further details on the geological history of the NAC are given in Supplementary Note 1). Combined with global zircon Hf datasets from other Mesoarchean to Neoarchean regions (Canada, Australia, southern Africa, South America), we demonstrate a widespread dilution of Hadean-to-Eoarchean crust in the Mesoarchean, coincident with other significant geochemical changes to our planet. These findings have implications for

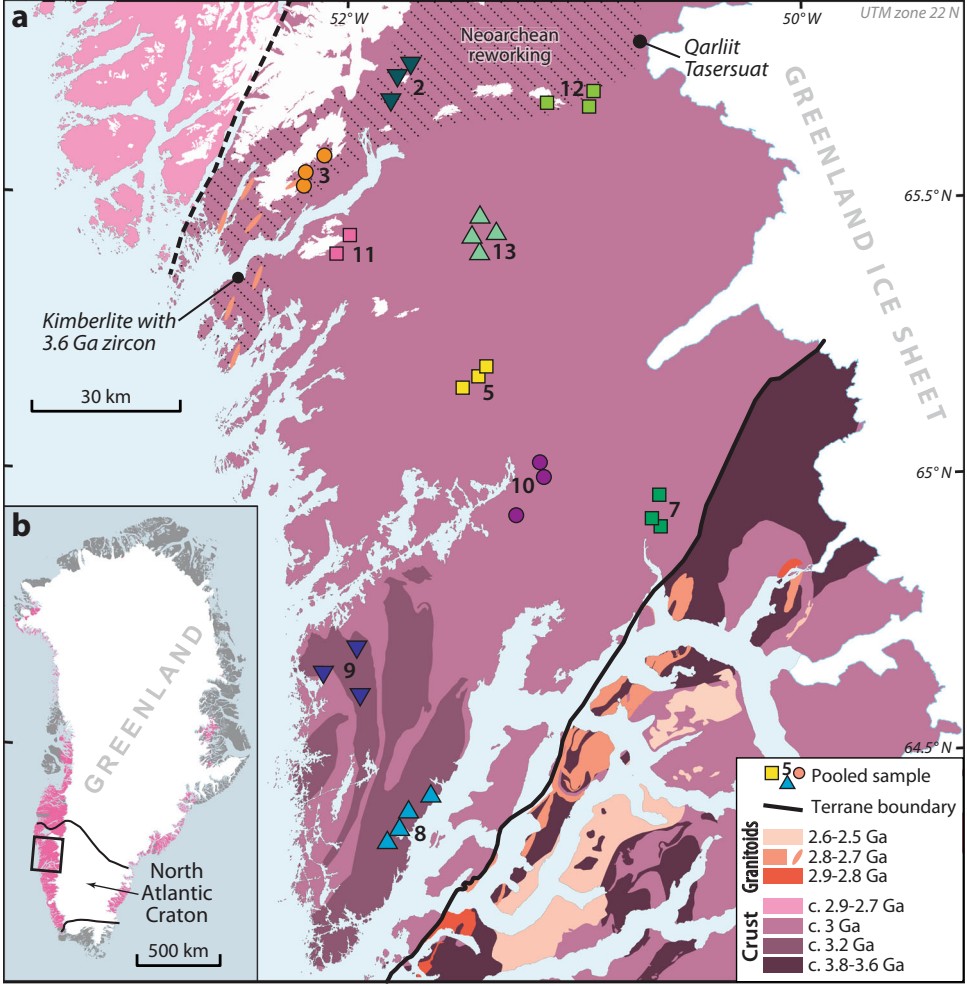

**Fig. 1 Lithological age map of the northern part of the North Atlantic Craton. a** Lithological age map of the northern part of the North Atlantic Craton showing stream sediment detrital zircon sample locations with sample identification number. Age map based on new field mapping and interpretations discussed in Friend and Nutman[67]. **b** Inset shows the location of the study area (black outlined box) with respect to exposures of Archean bedrock (pink hues) in Greenland.

models of early crustal evolution by illustrating the need for Hadean-to-Eoarchean precursors to preserve characteristic late Archean tonalite–trondhjemite–granodiorite (TTG).

## Results

**Provenance of stream sediment zircons.** Stream sediment was systematically sampled at a density of one sample station per 20–50 km$^2$ over large parts of Greenland[13]. Streams with catchment areas <20 km$^2$ were chosen to ensure that most material deposited at the sampling site was locally derived. Local derivation has been verified by study of the dispersal of indicator minerals from kimberlites in the surrounding surface environment[14]. The 0.1–1 mm grain size fraction of the stream sediment was used for zircon separation. A limited amount of each sample was available (30–50 g). Hence, to ensure a statistically robust number of zircon grains, ~120 g of material was produced by combining 2–4 individual samples from closely spaced catchment systems (Fig. 1). Ten resulting composited stream sediment samples were analyzed for U-Pb (Supplementary Data 1) and Lu-Hf isotopes (Supplementary Data 2). Reference material values are provided in Supplementary Data 3.

Most zircon grains have rounded terminations consistent with sedimentary transport. Cathodoluminescence (CL) images are dominated by textures characteristic of primary magmatic genesis including oscillatory zoning (Fig. 2 and Supplementary Data 4).

Metamorphic features including homogeneous rims are present on some grains but analyses targeted zones of primary magmatic texture. Most analyses (>94%) are within 10% discordance limits (Supplementary Note 2). The detrital dataset shows major age peaks at c. 3.2, 3.0 and 2.8–2.7 Ga, with minor components as young as 1.8 Ga and as old as c. 3.8 Ga (Fig. 3 and Supplementary Fig. 1). The c. 3.0 Ga detrital zircon age component is ubiquitous throughout the stream sediment samples and directly corresponds to the dominant age of crystalline rock within the local region[15,16] (Fig. 1). A 2.8–2.7 Ga detrital component accounts for a significant percentage of some samples (2, 3, 12). Regional terranes in the NAC have gneisses with zircon components in this age range[16,17]. A minor age component at 3.2 Ga in samples 8 and 9 is consistent with the age of diorite in the southern part of the area (Fig. 1; ref. [18]).

There is a significant proportion of 3.8–3.6 Ga detritus in one sample (~9%, sample 12), with other samples also showing minor proportions of Eoarchean-to-Paleoarchean components (Fig. 3 and Supplementary Table 1). Zircon grains >3.2 Ga are rare in the Mesoarchean part of the northern NAC; understanding the nature of this detritus has significant implications for the earliest history and construction of the NAC. The restricted catchment that these samples were acquired from implies a source endemic to the NAC. Dated crystalline rocks 25 km east of sample 12, at Qarliit Tasersuat, contain similar Neoarchean and Eo-Palaeoarchean zircon age components[19,20]. The most feasible

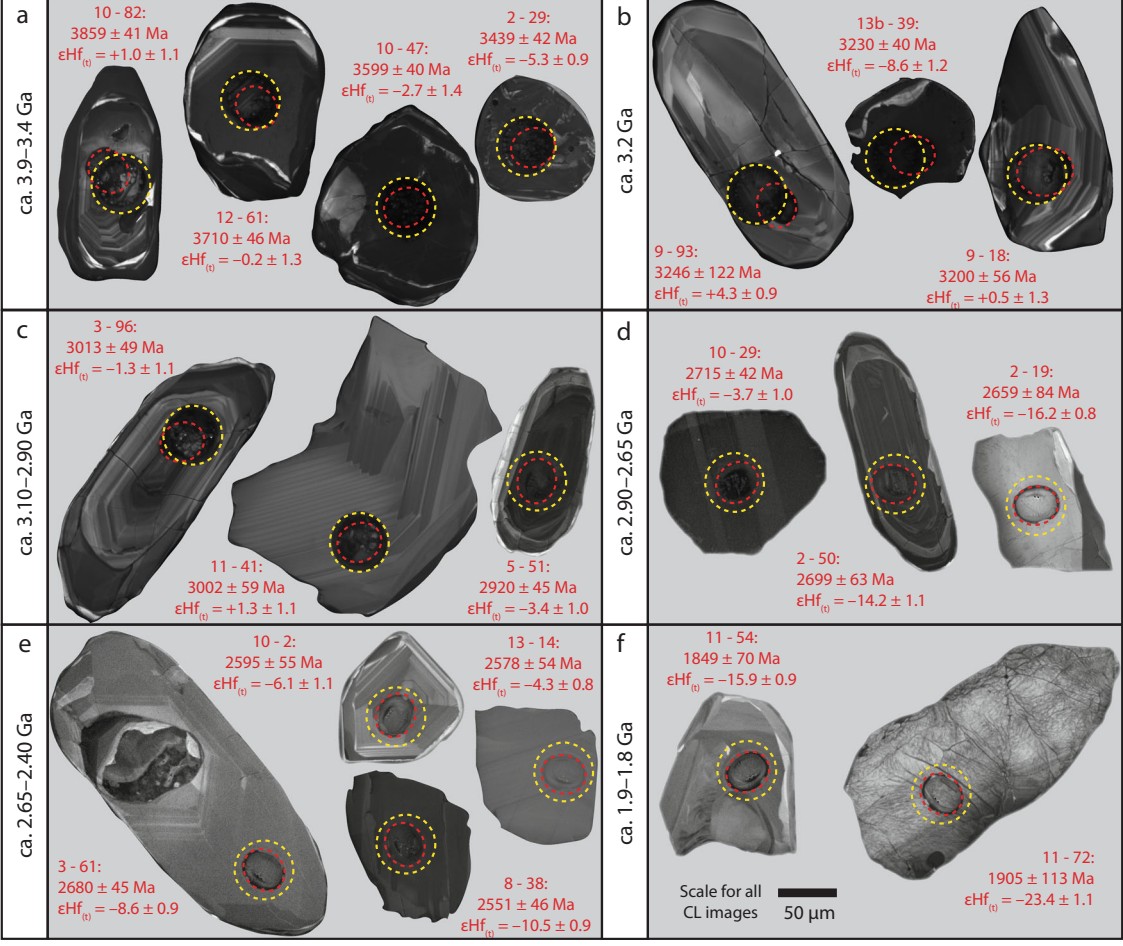

**Fig. 2 Cathodoluminescence images of detrital zircon grains.** Representative cathodoluminescence images of detrital zircon grains from stream sediments of the northern North Atlantic Craton. Panels show zircon of **a** ca. 3.90–3.40 Ga, **b** ca. 3.20 Ga, **c** ca. 3.10–2.90 Ga, **d** ca. 2.90–2.65 Ga, **e** ca. 2.65–2.40 Ga, and **f** ca. 1.90–1.80 Ga age. Red and yellow circles denote U-Pb and Lu-Hf analytical sites, respectively. Images of all grains are provided in Supplementary Data 4.

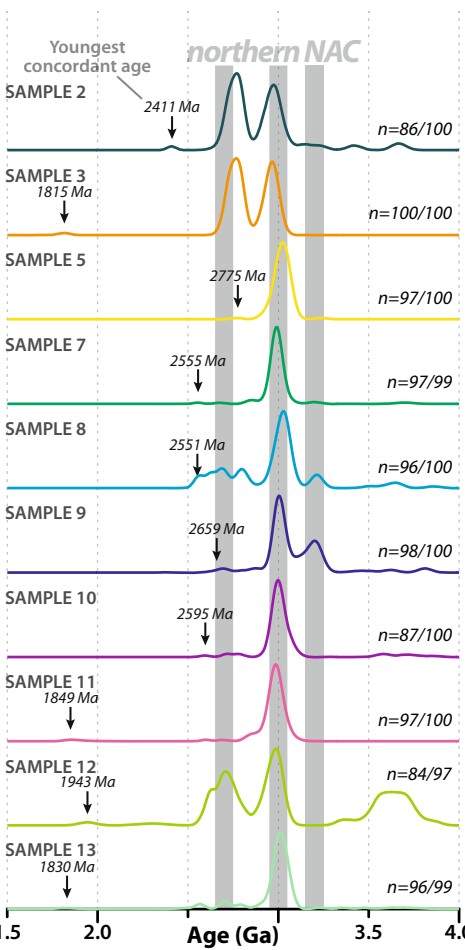

**Fig. 3 Probability density diagrams of zircon U-Pb dates.** Stacked probability density diagrams of zircon dates from stream sediment samples in the northern North Atlantic Craton. All ages are $^{207}$Pb/$^{206}$Pb and <10% discordant. The number of analyses within concordant limits versus all analyses is denoted with 'n'. The major age peaks for magmatic rocks in the northern NAC are indicated by gray vertical bars. Source data information for this figure is provided in Supplementary Data 1.

mechanism to source >3.2 Ga zircon crystals is to incorporate them as either xenocrysts or from xenoliths in later magmas that assimilated this material on transit through the basement. Alternatively, erosion out of younger (meta)sedimentary slivers could provide this material; however, no known (meta)sedimentary rocks are present in the catchments nor does the age spectra match that of ≤2877 million year old (Ma) and ≥2857 Ma supracrustal rocks[21].

**Hadean-to-Eoarchean crustal evolution in the northern North Atlantic Craton.** Hf isotopic values in zircon crystals reflect the degree to which the magma source was derived from the mantle or the crust. Hence, deflection of secular isotopic trends toward suprachrondritic values may imply, on average, greater asthenospheric input. In contrast, trends toward subchrondritic values imply greater crustal reprocessing. Hafnium isotopic data show that each of the main magmatic episodes in the northern NAC sourced a (moderate to minor) Hadean-to-Eoarchean component (Supplementary Note 1 and Fig. 4). The continuous, albeit limited, tapping of this source implies that it was consistently available and served as a basement component for all later magmas in this region.

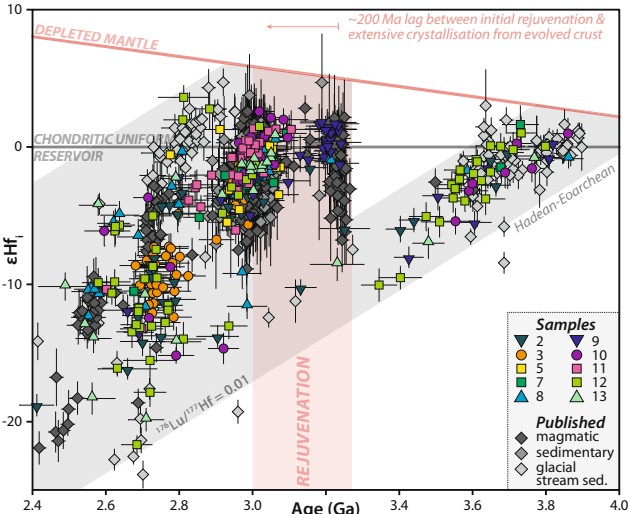

**Fig. 4 Zircon εHf isotope evolution diagram.** Zircon εHf isotope evolution diagram showing that the most evolved compositions at any given time period correspond to a Hadean-to-Eoarchean crustal source. Note major addition of juvenile material at c. 3.2 Ga. Depleted mantle values: $^{176}$Lu/$^{177}$Hf = 0.0388, $^{176}$Hf/$^{177}$Hf = 0.28325[68]. CHUR values: $^{176}$Lu/$^{177}$Hf = 0.0336, $^{176}$Hf/$^{177}$Hf = 0.282785[65]. $\lambda^{176}$Lu = 1.867 × 10$^{-11}$ a$^{-1}$ [64]. Published datasets are from Gardiner et al.[17] and Yi et al.[20]. Shaded field highlights limits of the dataset in this work. Error bars are shown at two standard errors. Source data information for this figure is provided in Supplementary Data 2.

Gardiner et al.[17] argued, on the basis of a lack of pre-3.2 Ga xenocrystic zircon or similarly old magmatic bodies, that the 3.2–2.6 Ga felsic crust of this region was generated from mafic crust lacking an Eoarchean felsic component. However, the now apparent vertical εHf$_i$ mixing trends for magmatism at 3.2, 3.0, and 2.75 Ga all point to a source component rooted in the Hadean-to-Eoarchean array (Fig. 4). While such an ancient component may only reflect a minor and presumably digested contribution to some episodes of magmatism (e.g., 3.0 Ga), it appears consistently. The highly evolved (negative) Hf signal is especially prominent during 2.75–2.70 Ga magmatism and was the dominant isotopic component sourced in 2.5–2.4 Ga zircon crystals entrained in a kimberlite (Fig. 4). Furthermore, the $^{176}$Lu/$^{177}$Hf slope of ~0.01 is consistent with an upper continental crust signature[12]. The implication of these observations is that, rather than under-thrusting of an ancient component at 3.0 Ga[22], this old source component was at least in place by 3.2 Ga. Furthermore, given the consistency of this ancient source component, a more likely situation is that a Hadean-to-Eoarchean felsic framework existed as the substrate through which the younger crust grew. This substrate interacted to various degrees with a more depleted reservoir, which in the 3.2–3.0 Ga period appears to be only slightly suprachondritic. The Hf isotopic signature in icecap melt-water sands south of the study area (near Isua) also reveals a comparable evolution pattern with components within the now well-defined Hadean-to-Eoarchean array[20].

## Discussion
While most zircon-bearing granitic magmas will reflect mixtures from different sources, such variability in isotopic signal over time carries important geodynamic information[23]. Specifically, different percentiles of Hf evolution plots have been regarded as principally carrying information about distinct geological

processes. For example, the 95th percentile has been regarded as tracking secular changes in mantle (juvenile) input, related in part to the supercontinent cycle[24]. To evaluate the degree of ancient basement recycling through time, the 5th percentile was tracked on a moving 50 million year (Myr) bin through our NAC εHf$_i$ data set. The 5th percentile is least contaminated by any mantle source and effectively tracks reworking of the Hadean-to-Eoarchean component in the study region (Fig. 5).

The Hadean-to-Eoarchean signal from southwest Greenland is compared to the Hf isotopic trend in global Archean terranes over the same 4.0–2.5 Ga period. We use detrended (i.e., subtraction of a linear regression line from the binned Hf values; e.g., ref. [25]) locally weighted scatterplot smoothing[26] curves with bootstrapped uncertainties that allow interrogation of the second-order variations (i.e., excluding first-order planetary differentiation, e.g., ref. [27]). Bins with less than five data points are considered to have too few data points to be robust and are excluded from consideration, though their inclusion results in no significant difference to the fit. These fits to Hf compilations from Canada, Australia, southern Africa, South America, and Greenland (all published data from ref. [28], with new Greenland data from this study) reveal a distinct pulse in juvenile magma into primordial crust between 3.2 and 3.0 Ga in all data sets (Fig. 6). Enhanced mantle input, rather than greater crustal reworking, is argued by suprachondritic values in 3.2 Ga zircon from all regions, which demand new juvenile addition from at least <3.5 Ga (Figs. 5 and 6). Hence, there is a global signal of increased juvenile input, diluting primordial Hadean-to-Eoarchean crust at the Paleoarchean-to-Mesoarchean boundary. This global signal implies an underlying synchronous process.

Our results show a general temporal consistency between a surge in mantle input into pre-existing crust and 3.2–3.0 Ga peaks in MgO, Ni, and Cr in basalts (Fig. 6; refs. [29,30]). The peak in juvenile addition also corresponds with a maximum in new crust formation ages[5] and an inflection in mantle depletion curves recorded from Nb/Th ratios in melts[31] and Os model ages[32]. These changes coincide with the petrological estimate of maximum mantle potential temperature[33] and the maximum planetary mantle temperature in secular cooling models[34–36]. A Urey ratio (mantle heat production divided by heat loss) of c. 0.23 is consistent with a thermal peak at the same time as geochemical proxies of maximum mantle temperature. Peak inflexion in the 5th percentile in Hf values slightly lags geochemical proxies of mantle addition in basalts[30]. This delay implies that crustal processing could have taken up to 200 Myr to respond to peak mantle temperatures, consistent with the zircon Hf record reflecting crust–mantle mixtures with an ancient recycled component (Fig. 7). Such a voluminous and widespread crustal reworking event requires an appropriately large thermal driver. We contend that all these observations can be rationalized with generation of enhanced mantle melt fractions at 3.2–3.0 Ga due to elevated mantle convective geotherms (Fig. 7a; ref. [37]).

During the Mesoarchean, mantle potential temperatures are thought to have been ~250–350 °C hotter than the present-day value of ~1350 °C[33,38]. Although the magnitude of such difference is poorly constrained, thermodynamic models for dry melting of mantle peridotite in the Mesoarchean predict greater depth/pressure of initial melting—intersection of adiabat with the solidus—and generation of greater mantle melt fractions overall than present day, potentially in excess of 40% if significant melting occurs at low pressures (Fig. 7a; ref. [39]). Similar arguments have been made based on the composition of komatiite magmas produced during this period[40].

Higher mantle temperatures in the Archean have also been considered to preclude or limit stable subduction, requiring a transition to dominant plate tectonics from another tectonic

mode (e.g., ref. [41]). How mantle temperature may have influenced crust production and recycling can be tracked using secular changes in isotope systems from magmatic crystals (e.g., zircon Hf). Thus the shift to more juvenile Hf isotopic values of magmatic rocks globally during this time is inferred to reflect

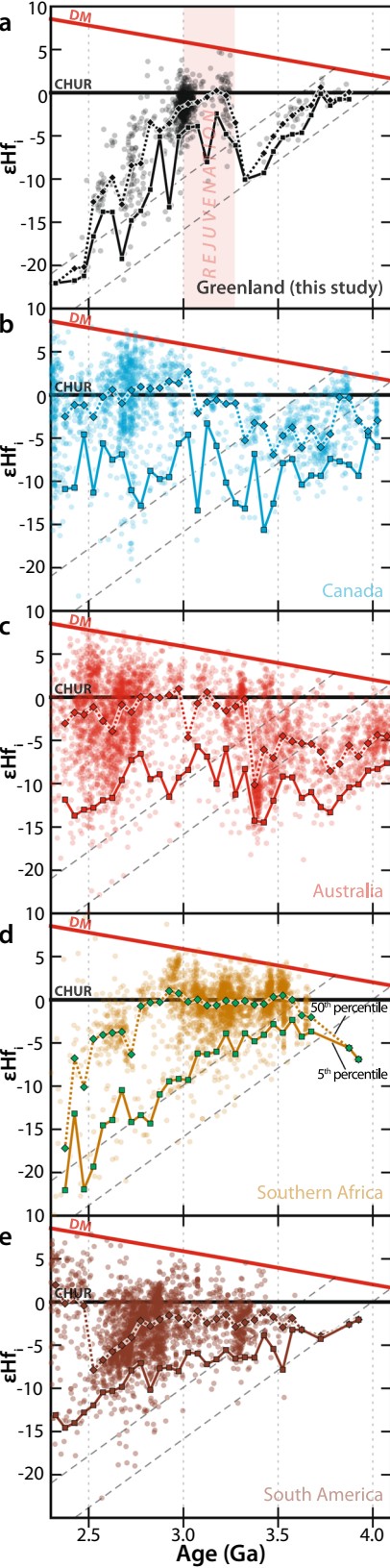

**Fig. 5 Epsilon Hf evolution diagrams for different cratonic regions.**
Epsilon Hf evolution diagrams for zircon analyses within different cratonic regions. Regions **a** Greenland, **b** Canada, **c** Australia, **d** Southern Africa, **e** South America. 5th (solid) and 50th (dashed) percentile fits, on a moving 50 Myr bin, are shown. Dashed gray lines denote average crustal evolution from a Hadean-to-Eoarchean source with $^{176}Lu/^{177}Hf$ of 0.01. DM depleted mantle, CHUR chondritic uniform reservoir. Source data information is provided in Supplementary Data 5 and Puetz et al.[28].

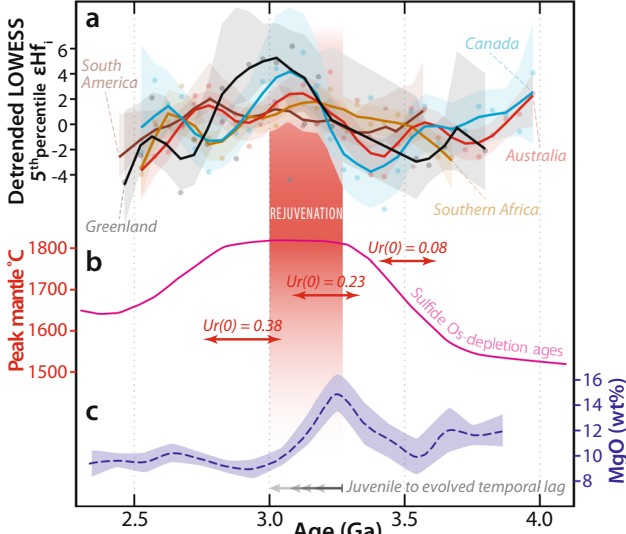

**Fig. 6 Global Hf evolution compared to geochemical proxies of mantle condition. a** Global compilation[28] of detrended LOWESS (locally weighted scatterplot smoothing) fits to the 5th percentile εHf$_i$ for the different Archean cratons in Fig. 5. Fits include faded 95% confidence band. **b** Sulfide Os-depletion model age probability density plot from Pearson et al.[32] and peak mantle temperatures for different Urey ratios (Ur) shown as red arrows[36]. **c** Global mean MgO in basalts from El Dien et al.[30], shown with two standard error of the mean uncertainties determined after bootstrap resampling. The concomitant increase in the 5th percentile εHf$_i$ and geochemical parameters of mantle temperature indicate a global crustal rejuvenation event at c. 3.2–3.0 Ga. Source data information is provided in Supplementary Data 5.

formation and injection of hot mafic/ultramafic magmas into pre-existing Hadean-to-Eoarchean crust (Fig. 7b).

Ancient continental crust produced in the early Earth thermal environment is mostly composed of variably deformed and metamorphosed magmatic rocks of the TTG suite[42] that formed by partial melting of older hydrated mafic rocks[43]. Generally, two stages of mantle differentiation are inferred to generate continental crust[44]. First, basaltic magma is extracted from the mantle. Second, it is buried and partially melted to form felsic continental crust. Global Hf isotopic patterns imply that early stage fractionated and extracted buoyant crust was re-fertilized via injection of mantle-derived magma during Earth's thermal peak. Such multiple stage crust production is consistent with results from elsewhere that support TTG generation from more ancient nuclei[6,45,46].

Smithies, et al.[6] linked the generation of stable tracts of continental crust to processes that produced mantle enrichments, envisaging the development of enriched mafic source regions forming Paleoarchean TTG at a particular stage (pre c. 3.5 Ga) in the compositional differentiation of the mantle. This work indicates another important later stage in TTG production (c. 3.2–3.0 Ga) during injection of mantle-derived material into this pre-existing buoyant crust.

Given the paucity of preserved Hadean-to-Eoarchean material in Archean crust today, the flux of juvenile material during the peak in mantle potential temperatures was sufficient to substantially rework primordial felsic crust. Buoyant felsic Hadean-to-Eoarchean material need not have made up a large volume of early crust but locally must have been sufficient to stall hot mafic–ultramafic magma and ultimately generate younger TTGs. Much Archean crust today may have only survived because it was emplaced into older Hadean-to-Eoarchean buoyant crust, which served as a life raft to preserve our planet's formative crust through antiquity.

## Methods

**Isotopic analysis of zircon.** U-Pb and Lu-Hf measurements were performed using in situ analytical techniques on zircon separates. Zircon grains were separated from each sample using magnetic and heavy liquid techniques. The zircon grains, together with zircon reference standards, were cast in 25 mm epoxy mounts, which were then polished to approximately half-grain thickness for analysis. Each mount was documented with transmitted and reflected light micrograph images prior to analysis. Post analysis, CL imaging was performed using a Tescan Mira3 FEG-SEM at the John de Laeter Centre, Curtin University to verify the spot positioning and provide information on internal textures to aid further interpretation.

U-Pb geochronology and Lu-Hf isotope analyses were performed at the John de Laeter Centre, Curtin University. U-Pb analyses were collected across two analytical sessions using a ~20-µm spot. Lu-Hf analyses were subsequently collected in a single session with a ~50-µm spot placed on top of the U-Pb spot on grains where the internal texture and size permitted the analysis of the same growth domain as best could be determined from imaging. For both U-Pb sessions, the excimer laser (RESOlution LR 193 nm ArF with a Lauren Technic S155 cell) on-sample energy was 2.0 J cm$^{-2}$ with a repetition rate of 5 Hz for 45 s of analysis time and ~60 s of background capture. All analyses were preceded by three cleaning pulses. For Hf analysis, the same laser was used with an on-sample energy of 3.0 J cm$^{-2}$ and a repetition rate of 10 Hz for 30 s of analysis time and ~60 s of background capture. For both U-Pb and Hf, the sample cell was flushed by ultrahigh purity He (320 mL min$^{-1}$) and N$_2$ (2.8 mL min$^{-1}$).

For both sessions, U-Pb data were collected on an Agilent 8900 triple quadrupole mass spectrometer with high-purity Ar as the carrier gas for both sessions (flow rate 0.98 L min$^{-1}$). Analyses of 10 unknowns were bracketed by analysis of a standard block containing the primary zircon reference materials GJ-1 (601.86 ± 0.37 Ma; refs. [47,48]) and OG1 (3465.4 ± 0.6 Ma; ref. [49]), which were used to monitor and correct for mass fractionation and instrumental drift for U-Pb dating. Secondary reference zircon were used to monitor data accuracy and precision, including Plešovice (337.13 ± 0.37 Ma; ref. [50]), 91500 (1062.4 ± 0.4 Ma; ref. [51]), and Maniitsoq (3008.70 ± 0.72 Ma; ref. [52]), all uncertainties at 2σ. During the analytical sessions, when reduced against a matrix-matched reference material, Plešovice, 91500, and Maniitsoq yielded statistically reliable ($p > 0.05$) weighted mean ages of 338.2 ± 0.8–338.6 ± 0.9, 1062.0 ± 3.9–1063.4 ± 2.7, and 3003 ± 10–3010 ± 10 Ma, respectively, all of which are within 2σ of the published age (Supplementary Data 3 provides full U-Pb standard compilation). For this study, the primary reference material was zircon OG1 (3465.4 ± 0.6 Ma; ref. [49]), which is an Archean zircon with similar ablation response to the unknowns investigated in this work.

Time-resolved mass spectra were reduced using the U/Pb Geochronology3 reduction schemes in Iolite$^{TM}$ [53] and in-house Microsoft Excel macros. No common lead corrections were deemed necessary due to low $^{204}Pb$ counts; the proportion of common Pb ($f$206) was <0.05% or below detection limit for 983 of 995 analyses, with the remaining 12 analyses between 0.05 and 0.15%. All dates are reported with ±2σ uncertainties calculated directly from isotopic ratios. Uncertainties on the primary reference materials were propagated in quadrature to the unknowns and secondary zircon reference materials. Age calculations and inverse concordia plots utilized the Isoplot 4.15 software[54]. Kernel density plots of detrital zircon age populations were produced in the R statistical "provenance" analysis package[55]. Full isotopic data for the samples are given in Supplementary Data 1 and values for the reference materials are given in Supplementary Data 3. All zircon ages are presented as $^{207}Pb/^{206}Pb$ ages due to their superior precision at ages between 4.0 and 1.8 Ga.

Lu–Hf isotopic data (Supplementary Data 2) were collected on a Nu Instruments Plasma II multi-collector inductively coupled plasma mass spectrometry (ICP-MS). Measurements of $^{171}Yb$ $^{172}Yb$, $^{173}Yb$, $^{175}Lu$, $^{176}Hf + Yb$ +Lu, $^{177}Hf$, $^{178}Hf$, $^{179}Hf$, and $^{180}Hf$ were made simultaneously. Time-resolved data were baseline subtracted and reduced using Iolite data reduction scheme after ref. [56], where $^{176}Yb$ and $^{176}Lu$ were removed from the 176 mass signal using $^{176}Yb/$ $^{173}Yb = 0.7962$ and $^{176}Lu/^{175}Lu = 0.02655$ with an exponential law mass bias correction assuming $^{172}Yb/^{173}Yb = 1.35274$[57]. The interference corrected $^{176}Hf/$ $^{177}Hf$ was normalized assuming $^{179}Hf/^{177}Hf = 0.7325$[58] for mass bias correction. The mass spectrometer was initially tuned using Yb- and Lu-doped JMC475 Hf solutions introduced using the Aridus II desolvating nebulizer, with the goal of maximizing Hf detection efficiency while minimizing oxide production. A range of different zircon crystals from reference standards were analyzed together with the samples in each session to monitor the accuracy of the results. Mud Tank zircon

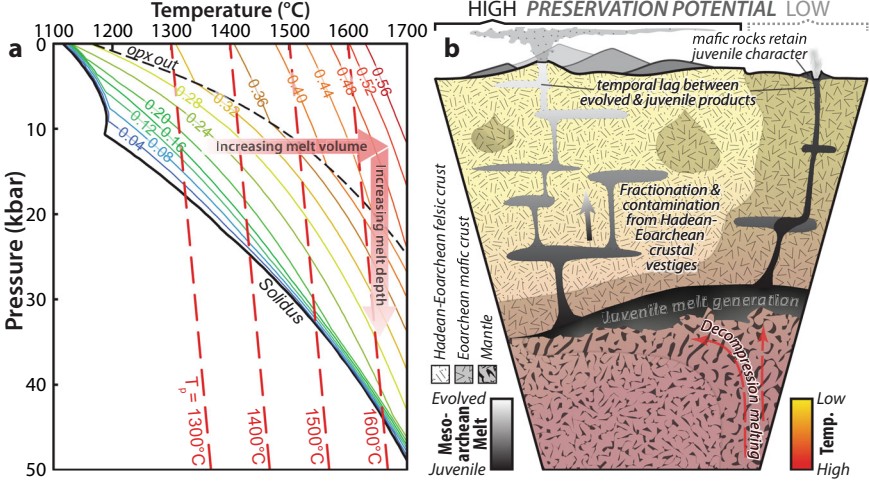

**Fig. 7 Pressure–temperature diagram depicting the melting of mantle peridotite and a schematic representation of the implication of this process for ancient crust. a** Pressure–temperature diagram showing location of the solidus and various melt isopleths during dry melting of mantle peridotite modified after ref. [39], along with adiabatic convective geotherms for mantle potential temperatures of 1300, 1400, 1500, and 1600 °C (after McKenzie and O'Nions[69]). Overall, increasing mantle potential temperatures result in greater melting depth and greater melt fractions. Opx denotes Orthopyroxene. **b** Cartoon section depicting the production of Mesoarchean mantle-derived melts, which penetrate and modify pre-existing crust and generate new TTG (tonalite–trondhjemite–granodiorite). Ancient crust serves as buoyant life-rafts that facilitate rapid crustal growth during the Mesoarchean and Neoarchean and are also preferentially preserved relative to solely juvenile segments.

was used as the primary reference material for Hf isotope ratios, with a $^{176}Hf/^{177}Hf$ ratio of $0.282507 \pm 0.000006$[59]. Corrected $^{176}Lu/^{177}Hf$ ratios were determined through processing against R33 zircon reference ($0.001989 \pm 0.000869$; ref. [60]). 91500 ($0.282305 \pm 0.000006$; ref. [61]), FC1 ($0.282184 \pm 0.000016$; ref. [59]), GJ-1 ($0.282000 \pm 0.000009$; ref. [62]), and Plešovice ($0.282474 \pm 0.000012$; ref. [50]) were used as secondary references to monitor accuracy of data processing. During the analytical session, all analyzed secondary references had statistically significant means (i.e., $p > 0.05$) that fell within $2\sigma$ error of reported $^{176}Hf/^{177}Hf$ values (Supplementary Data 3), including 91500 ($0.282288 \pm 0.00011$), FC1 ($0.282172 \pm 0.000009$), GJ-1 ($0.282011 \pm 0.000009$), and Plešovice ($0.282474 \pm 0.000013$). The stable $^{178}Hf/^{177}Hf$ and $^{180}Hf/^{177}Hf$ ratios for Mud Tank yielded values of $1.467201 \pm 0.000011$ and $1.886829 \pm 0.000014$, respectively, which overlap at $2\sigma$ with recommended values reported by Thirlwall and Anczkiewicz[63] (Supplementary Data 3). Decay constants, chondritic uniform reservoir (CHUR), and depleted mantle values are taken from Söderlund et al.[64], Bouvier et al.[65], and Griffin et al.[66], respectively. Epsilon notation Hf isotopic data relative to CHUR at the initial time of crystallization $[\varepsilon Hf_i]$ were computed for concordant zircon grains using the age of the crystal, and the measured (present-day) $^{176}Lu/^{177}Hf$ and $^{176}Hf/^{177}Hf$ values. Elevated Yb content may have a deleterious effect on isobaric interference correction of $^{176}Hf/^{177}Hf$. However, the median Yb content of stream sediment zircon is 2200 ppm on $^{176}Hf$, which is less than that in reference material FC1. FC1 reproduces the expected $^{176}Hf/^{177}Hf$ ratio yet has, on average, much greater Yb interference (4000 ppm on $^{176}Hf$) than the sample material.

## Data availability

All data used in this manuscript are included in the Supplementary Information File and Supplementary Data Files (1–5).

## Code availability

Iolite used for U/Pb and Lu/Hf data reduction is available through https://iolite.xyz/.

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

## Acknowledgements

GeoHistory Facility instruments in the John de Laeter Centre, Curtin University were funded via an Australian Geophysical Observing System grant provided to AuScope Pty Ltd by the AQ44 Australian Education Investment Fund program. The NPII multi-collector was obtained via funding from the Australian Research Council LIEF program (LE150100013). Brad McDonald and Noreen Evans are thanked for support during LA-ICP-MS analysis. The Ministry of Minerals and Resources, Government of Greenland funded the sample processing and analysis.

## Author contributions

C.L.K., A.S., and J.A.H. designed the study. A.S. provided the samples and C.L.K. collected and processed the data, M.I.H.H., H.K.H.O., and M.B. contributed further data reduction and analysis. M.B. and H.K.H.O. drafted the figures. C.L.K. led the writing of the manuscript with contributions from all authors.

## Competing interests

The authors declare no competing interests.
