## [Peer Review File · Nature Communications]

REVIEWER COMMENTS

Reviewer #1 (Remarks to the Author):

The manuscript reports the results of Hf isotopic analyses of 3.9 to 1.8 billion year-old new detrital zircon (dated by U-Pb) from stream sediment samples from West Greenland. The results suggest reworking of felsic Hadean-Eoarchean crust during later magmatic events. The detrital West Greenland zircon data were then compared with global Hf datasets and show that a shift to more juvenile Hf values occurs between 3.0 and 3.2 billion years ago. The authors point to a thermal peak at this time which produced a higher fraction of melting of the mantle and injecting into older Hadean-Eoarchean felsic crust. This process may have also protected these earliest Earth's crustal nuclei during subsequent crustal growth.

The work contains high quality data from a new selection of West Greenland detrital zircons and a robust analysis of the data sets. The comparison with global Hf zircon data show a correlation that can provide some evidence for higher potential mantle temperatures in the Archean, an idea that is debated in the community. The link between the datasets and higher potential temperatures of the Archean mantle of 250-300oC hotter are described in the Discussion section (lines 136-145). However, these implications could be developed further and be a more prominent part of the paper. The authors could address some of the debate about a warmer mantle in the Archean. Discussion lines 110 to 120 are somewhat difficult to access and I recommend a more general description of the results.

In the conclusions and in terms of general application, the results presented here are of most interest to those in the field of early Earth research. I think References give appropriate credit to earlier research and the quality of the figures and the clarity of the abstract, summary and overall presentation is very high.

Reviewer #2 (Remarks to the Author):

I was delighted to read the manuscript entitled "Widespread reworking of Hadean to Eoarchean continents during Earth's thermal peak by Kirkland et al. I find this manuscript easy to read, well-structured and of broad interest for the geoscience community. The introduction is of broad interest and set the stage well for the data presentation. The data is sound and supports the interpretations. The interpretation of the data set is clear. I would fully recommend the manuscript for publication; however some minor points need clarification. I hope these points will help to broaden the readership of the paper.

The first observations that stroke me is the fact that magmatic episodes in the northern NAC sourced some sort of Eoarchean and Hadean crust. This is a feature that I have long noticed working in South America and Southern Africa. This really strikes me because most models for TTG generation back then (10-15 years ago) did require juvenile island arc terrains. So, this peak in juvenile-magma addition in the NAC or elsewhere should be explained in view of early and most recent models for TTG generation.

The authors made it pretty convincing that there is a period of major magmatic addition between 3 and 3.2, which coincides with maximum mantle potential. I agree with them in this point, but how does the juvenile magma reconcile current models for Archean crust generation? I would suggest exploring figure 6 a bit more and explain their model in view of the current models for Archean juvenile crust formation.

In the introduction (lines 54-55) the authors made a point in which models for evolution of early crust have been tested. I did not see it clearly in the discussion, and I think the authors should explain a bit more the nature of this magma at 3.2 Ga.

The authors claim that the zircons analyzed in their study are dominantly magmatic. I find it

difficult to understand the lack of metamorphic zircons, given the age and reworked nature of the crust that authors report. Why such sediments did not sample metamorphic zircons? In my experience here in South America, India and Africa shows that nearly 50% of Archean zircons record some sort of metamorphic feature, like metamorphic rims. If so, how representative is this sampling site? Can it be used as a proxy for a global Archean crust?

I fully agree with the authors conclusions of a preexisting Hadean crust that hosted mantle derived and crustal derived felsic magmas. In fact, my thoughts about the origin of most Archean zircons are clearly expressed in this paper, as nearly 50% of the late Archean detrital zircons that I have dated is generally crustal. Even some oldest and best characterized TTG rocks in South America or South Africa seem to record some sort of crustal contamination. However, I was a bit disappointed with the fact that the authors did not care to look into a broader dataset of Archean zircons. There are plenty of Archean-Hadean detrital zircon datasets in South America for instance that could be used to complement this model. Also, there are plenty of Sm-Nd data that could support the Hf dataset into a broader geodynamic model.

My point is that the authors should explore this impressive evolution curve for the Hadean crust. The paper would be a lot stronger if the authors combine the world Hf zircon dataset in one diagram that is focused on the Hadean crustal evolution. The diagrams shown in Figure 5 are small blurry and difficult to read. The focus of this paper is the Eoarchean and Hadean crust, so re-scale the diagrams so the readers can focus on that.

I found the methods description adequate. However, I would like to see a bit more detail on the Hf methodology used for this paper. For instance, these zircons are Archean and probably enriched in REE and particularly Yb. It is important that the authors describe a bit more about the Yb corrections, and % of Yb corrections necessary for measuring the Hf ratios correctly.

I find that the diagrams are blurry and difficult to read. I think they should explore a bit more the diagrams, pinning down their conclusions more clearly with these diagrams. If there is a problem with manuscript size, I would recommend putting figure 2 in the supplement and redrawing figure 3 so that you have more space for the diagrams. There is too much text about the U-Pb ages, which is hardly used in the discussion. This could also be reduced or put in the supplement.

I think the point of this paper is brilliant, but it should be better explained. The authors should focus on what matters, which is the Hf signatures of the NAC and of the Archean rocks around the world.

I find it very difficult to see the difference in juvenile addition at 3.2 3.0 and 2.8 Ga in Fig. 4. The symbols are too big and lines are too thin.

I have not seen a single spelling mistake, but I presume this should be picked up by the editorial office. I have not seen a single problem with the reference list and with the supplementary data. Finally, I would love to see a more detailed cartoon illustrating how this juvenile crust addition reconcile with models for TTG crust formation. It would be very useful.

Response letter NCOMMS-20-31507A

We sincerely thank both reviewers for their comments. We respond to all comments below, indicating our changes to the manuscript in light of these reviews.

Responses to reviewer's comments are provided in blue font. Text edits to the manuscript are indicated in **bold** and text quoted from the paper are in *italics*.

REVIEWER COMMENTS / RESPONSES

Reviewer #1 (Remarks to the Author):

The manuscript reports the results of Hf isotopic analyses of 3.9 to 1.8 billion year-old new detrital zircon (dated by U-Pb) from stream sediment samples from West Greenland. The results suggest reworking of felsic Hadean-Eoarchean crust during later magmatic events. The detrital West Greenland zircon data were then compared with global Hf datasets and show that a shift to more juvenile Hf values occurs between 3.0 and 3.2 billion years ago. The authors point to a thermal peak at this time which produced a higher fraction of melting of the mantle and injecting into older Hadean-Eoarchean felsic crust. This process may have also protected these earliest Earth's crustal nuclei during subsequent crustal growth.

The work contains high quality data from a new selection of West Greenland detrital zircons and a robust analysis of the data sets. The comparison with global Hf zircon data show a correlation that can provide some evidence for higher potential mantle temperatures in the Archean, an idea that is debated in the community. The link between the datasets and higher potential temperatures of the Archean mantle of 250-300oC hotter are described in the Discussion section (lines 136-145).

However, these implications could be developed further and be a more prominent part of the paper. The authors could address some of the debate about a warmer mantle in the Archean.

In response to this comment on mantle temperatures in the Archean we modify the text to include further statements on this aspect:

“During the Mesoarchean, mantle potential temperatures are thought to have been ~250–350°C hotter than the present-day value of ~1350°C. Although the magnitude of such difference is poorly constrained, thermodynamic models for dry melting of mantle peridotite in the Mesoarchean predict greater depth/pressure of initial melting.....Higher mantle temperature in the Archean has been considered to preclude or limit stable subduction, requiring a transition to dominant plate tectonics from another tectonic mode. How mantle temperature may have influenced crust production and recycling may be tracked using secular changes in isotope systems from magmatic crystals.”

Discussion lines 110 to 120 are somewhat difficult to access and I recommend a more general description of the results.

Lines 110-120 are a brief description of our approach to find secular trends in isotopic datasets and the justification of the geological interpretation of such trends. We have attempted to make this segment of text more accessible and added additional references on the approach. However, we do need to retain some common statistical terms (percentile, detrended, LOWESS) as they have specific meaning that describes their use in quantifying the secular trends in the data.

In response to this comment we have modified the text to clarify the interpretations of the Hf isotope data trends:

Prior to the section interpreting Hf isotope data of the NAC we add “Hf isotopic values in zircon crystals reflect the degree to which the magma source was derived from the mantle or the crust. Hence, deflection of secular isotopic trends towards supra chondritic values may imply, on average, greater asthenospheric input, in contrast trends towards sub chondritic values imply greater crustal reprocessing.”

*In the succeeding Discussion section we amend the text as follows “Whilst most zircon bearing granitic magmas will reflect mixtures from different sources, such variability in isotopic signal over time carries important geodynamic information (Kohanpour et al., 2018). Specifically, different percentiles of Hf evolution plots have been regarded as principally carrying information about distinct geological processes. For example, the 95th percentile has been regarded as tracking secular changes in mantle (juvenile) input, related in part to the supercontinent cycle²³. To evaluate the degree of ancient basement recycling through time the 5th percentile was tracked on a moving 50 Ma bin through our NAC ϵHf_t data set. The 5th percentile **is least contaminated by any mantle source and effectively tracks reworking of the Hadean-to-Eoarchean component in the study region (Fig. 5).***

*The Hadean-to-Eoarchean signal from southwest Greenland is compared to the Hf isotopic trend in global Archean terranes over the same 4.0–2.5 Ga period. We use detrended (i.e., **subtraction of a linear regression line from the binned Hf values; e.g. Van Kranendonk and Kirkland, 2016**) LOWESS (**Locally Weighted Scatterplot Smoothing; Cleveland, 1979**) curves with bootstrapped uncertainties that allow interrogation of the second-order variations (i.e., excluding first order planetary differentiation; e.g. **Brown et al., 2020**).”*

Kohanpour, F., Kirkland, C. L., Gorczyk, W., Occhipinti, S., Lindsay, M. D., Mole, D., & Le Vaillant, M. (2019). Hf isotopic fingerprinting of geodynamic settings: Integrating isotopes and numerical models. *Gondwana Research*, 73, 190-199 doi:<https://doi.org/10.1016/j.gr.2019.03.017>

Van Kranendonk, M.J., Kirkland, C.L. (2016): Conditioned duality of the Earth System: Geochemical tracing of the supercontinent cycle through the Precambrian. *Earth-Science Reviews* 160, 171-187. DOI [10.1016/j.earscirev.2016.05.009](https://doi.org/10.1016/j.earscirev.2016.05.009)

Cleveland, William S. (1979). Robust Locally Weighted Regression and Smoothing Scatterplots. *Journal of the American Statistical Association*. 74 (368): 829–836. doi:[10.2307/2286407](https://doi.org/10.2307/2286407). JSTOR 2286407. MR 0556476

Brown, M., Kirkland, C.L., & Johnson, T.E. (2020). Evolution of geodynamics since the Archean: Significant change at the dawn of the Phanerozoic. *Geology* doi:[10.1130/g47417.1](https://doi.org/10.1130/g47417.1)

In the conclusions and in terms of general application, the results presented here are of most interest to those in the field of early Earth research. I think References give appropriate credit to earlier research and the quality of the figures and the clarity of the abstract, summary and overall presentation is very high.

Reviewer #2 (Remarks to the Author):

I was delighted to read the manuscript entitled “Widespread reworking of Hadean to Eoarchean continents during Earths thermal peak by Kirkland et al. I find this manuscript easy to read, well-structured and of broad interest for the geoscience community. The introduction is of broad interest and set the stage well for the data presentation. The data is sound and supports the interpretations. The interpretation of the data set is clear. I would fully recommend the manuscript for publication; however some minor points need clarification. I hope these points will help to broaden the readership of the paper.

The first observations that stroke me is the fact that magmatic episodes in the northern NAC sourced some sort of Eoarchean and Hadean crust. This is a feature that I have long noticed working in South America and Southern Africa. This really strikes me because most models for TTG generation back then (10-15 years ago) did require juvenile island arc terrains. So, this peak in juvenile-magma addition in the NAC or elsewhere should be explained in view of early and most recent models for TTG generation.

TTG are widely regarded to form from hydrated mafic rocks and whilst Hf data alone does not provide unique insight into the definitive source compositions, it does provides a clear indication of the age of the precursors and the timing of new TTG generation, helping to define a mantle heat source for voluminous TTG production.

We modify the manuscript with the addition of the following text to better place TTG generation in the context of this work:

“Ancient continental crust produced in the early Earth thermal environment is mostly composed of variably deformed and metamorphosed magmatic rocks of the tonalite–trondhjemite–granodiorite (TTG) suite (Jahn et al., 1981) that formed by partial melting of older hydrated mafic rocks (Moyen & Martin, 2012). Generally, two stages of mantle differentiation are inferred to generate continental crust (Jain et al., 2019). First, basaltic magma is extracted from the mantle. Second, it is buried and partially melted to form felsic continental crust. Global Hf isotopic patterns imply that early stage fractionated and extracted buoyant crust was re-fertilised via injection of mantle derived magma during Earth’s thermal peak. Such multiple stage crust production is consistent with results from elsewhere that support TTG generation from more ancient nuclei (Smithies et al., 2009; Martin et al., 2014; Condie et al., 2018).”

Jahn, B.-M., Glikson, A.Y., Peucat, J.J., Hickman, A.H., 1981, REE geochemistry and isotopic data of Archean silicic volcanics and granitoids from the Pilbara Block, Western Australia: implications for the early crustal evolution. *Geochim. et Cosmochim. Acta* 45, 1633–1652.

Moyen, J.F., Martin, H., 2012, Forty years of TTG research. *Lithos*, 148, 211-223.

Jain, C., Rozel, A. B., Tackley, P. J., Sanan, P. Gerya, T.V., 2019, Growing primordial continental crust self-consistently in global mantle convection models. *Gondwana Research* 73, 96–122.

Smithies RH, Champion DC, Van Kranendonk MJ., 2009, Formation of Paleoarchean continental crust through infracrustal melting of enriched basalt. *Earth Planet. Sci. Lett.* 281:298–306

Martin H, Moyen JF, Guitreau M, Blichert-Toft J, Le Pennec JL., 2014, Why Archaean TTG cannot be generated by MORB melting in subduction zones. *Lithos* 198, 1–13

Condie KC, Puetz SJ, Davaille A. 2018, Episodic crustal production before 2.7 Ga. *Precambrian Res.* 312, 16–22.

The authors made it pretty convincing that there is a period of major magmatic addition between 3 and 3.2, which coincides with maximum mantle potential. I agree with them in this point, but how does the juvenile magma reconcile current models for Archean crust generation? I would suggest exploring figure 6 a bit more and explain their model in view of the current models for Archean juvenile crust formation.

We expand our discussion on TTG generation as per the proceeding response and also:

“Smithies et al., (2009) linked the generation of stable tracts of continental crust to processes that produced mantle enrichments, envisaging the development of enriched mafic source regions forming Paleoproterozoic TTG at a particular stage (pre c. 3.5 Ga) in the compositional differentiation of the mantle. This work indicates another important later stage in TTG production (c. 3.2–3.0 Ga) during injection of mantle derived material into this pre-existing buoyant crust.”

Fig. 6 has also been modified to visualise these points with respect to Archean crust generation.

In the introduction (lines 54-55) the authors made a point in which models for evolution of early crust have been tested. I did not see it clearly in the discussion, and I think the authors should explain a bit more the nature of this magma at 3.2 Ga.

To be more specific about the findings of this work we rephrase line 54-55 to:

“Combined with global zircon Hf datasets from other Meso- to Neoproterozoic regions (Canada, Australia, southern Africa, South America), we demonstrate a widespread dilution of Hadean-to-Proterozoic crust in the Proterozoic, coincident with other significant geochemical changes to our planet. These findings have implications for models of early crustal evolution by illustrating the need for Hadean-to-Proterozoic precursors to preserve late Archean TTG.”

The authors claim that the zircons analyzed in their study are dominantly magmatic. I find it difficult to understand the lack of metamorphic zircons, given the age and reworked nature of the crust that authors report. Why such sediments did not sample metamorphic zircons? In my experience here in South America, India and Africa shows that nearly 50% of Archean zircons record some sort of metamorphic feature, like metamorphic rims. If so, how representative is this sampling site? Can it be used as a proxy for a global Archean crust?

Metamorphic zircon tends to form rims on pre-existing magmatic zircon seeds (e.g. Rubatto, 2017). We provide CL imaging with textures indicative of magmatic zircon growth (in figure 2 and images of all grains are provided in the supplementary data). Whilst some grains do indeed have variable thicknesses of metamorphic overgrowth, we specifically targeted the primary oscillatory zoned domains, which preserve the magmatic source history. Furthermore, one would expect sedimentary processes to preferentially remove metamorphic rims to leave more of the primary magmatic cores of interest. We modify the text to discuss this aspect.

*“Most zircon grains have rounded terminations consistent with sedimentary transport. Cathodoluminescence images are dominated by textures characteristic of primary magmatic genesis including oscillatory zoning (Fig. 2). **Metamorphic features including homogeneous rims are present on some grains but analyses targeted zones of primary magmatic texture.**”*

The sampling sites can be judged as representative of the regional crustal substrate given the recovered ages (and their relative probabilities) well-match the regional crystal basement age structure (e.g. Gardiner et al., 2019). Furthermore, the similarity in signature from multiple samples provides confidence in the sampling approach. Comparison of the Hf signature in these stream sediment samples reveals temporal patterns that match global Archean isotope evolution patterns implying they accurately track crustal growth trends.

Rubatto, D. (2017). Zircon: The metamorphic mineral. *Reviews in Mineralogy and Geochemistry*, 83, 261–295.

Gardiner, N. J., Kirkland, C. L., Hollis, J., Szilas, K., Steenfelt, A., Yakymchuk, C., & Heide-Jørgensen, H. (2019). Building Mesoarchean crust upon Eoarchean roots: the Akia Terrane, West Greenland. *Contributions to Mineralogy and Petrology*, 174, 20 doi:<https://doi.org/10.1007/s00410-019-1554-x>

I fully agree with the authors conclusions of a preexisting Hadean crust that hosted mantle derived and crustal derived felsic magmas. In fact, my thoughts about the origin of most Archean zircons are clearly expressed in this paper, as nearly 50% of the late Archean detrital zircons that I have dated is generally crustal. Even some oldest and best characterized TTG rocks in South America or South Africa seem to record some sort of crustal contamination. However, I was a bit disappointed with the fact that the authors did not care to look into a broader dataset of Archean zircons. There are plenty of Archean-Hadean detrital zircon datasets in South America for instance that could be used to complement this model. Also, there are plenty of Sm-Nd data that could support the Hf dataset into a broader geodynamic model.

In response to this comment we include the fifth percentile curve generated from a compilation of Hf in South America. We are pleased to report data from this additional region also shows the same general influence of enhanced juvenile input during the planetary peak in mantle potential temperature.

While Sm/Nd could also be used to track crustal reprocessing versus juvenile addition in much the same way as zircon Hf, our preference is to use zircon Hf because such isotopic system can be related to the same directly dated material (zircon) and therefore provide a perhaps more nuanced temporal record. Furthermore, zircon is far more resistant to secondary processes. Lastly, we do not really want to dilute our narrative in this contribution by bringing in an additional isotope system that will, effectively, track similar processes.

My point is that the authors should explore this impressive evolution curve for the Hadean crust. The paper would be a lot stronger if the authors combine the world Hf zircon dataset in one diagram that is focused on the Hadean crustal evolution. The diagrams shown in Figure 5 are small blurry and difficult to read. The focus of this paper is the Eoarchean and Hadean crust, so re-scale the diagrams so the readers can focus on that.

Our review copy pdf down sampled the images, we trust the resolution of the images will be better in any future publication version. The scale on the figures is appropriate to contextualise the data we present and is centred on the 3.2 Ga juvenile addition, which is key to our thesis. As requested, we revise the Hf evolution figure and split it into two different figures with one showing the regional patterns and the second showing the combined detrended patterns with respect to other global signatures.

I found the methods description adequate. However, I would like to see a bit more detail on the Hf methodology used for this paper. For instance, these zircons are Archean and probably enriched in REE and particularly Yb. It is important that the authors describe a bit more about the Yb corrections, and % of Yb corrections necessary for measuring the Hf ratios correctly.

Elevated $^{176}\text{Yb}/^{177}\text{Hf}$ can lead to elevated epsilon Hf due to inefficient isobaric interference correction. However, we report five different Hf reference materials covering a range of Yb content and provide this information in the supplementary materials. All reference materials reproduce expected $^{176}\text{Hf}/^{177}\text{Hf}$ supporting the efficacy of our Yb correction routine. Furthermore, the median Yb content of unknown (stream sediment) zircon is 2200 ppm Yb on ^{176}Hf , which is much less than the Yb content (4000 ppm) of our high interference reference material FC1, which reproduces the expected $^{176}\text{Hf}/^{177}\text{Hf}$. Additionally, we see no correlation between $^{176}\text{Hf}/^{177}\text{Hf}$ and Yb content in our

data set, an observation that provides no evidence for ineffective interference correction. In response to this comment we modify the method text to include the following:

“Elevated Yb content may have a deleterious effect on isobaric interference correction of $^{176}\text{Hf}/^{177}\text{Hf}$. However, the median Yb content of stream sediment zircon is 2200 ppm on ^{176}Hf , which is less than that in reference material FC1. FC1 reproduces the expected $^{176}\text{Hf}/^{177}\text{Hf}$ ratio yet has, on average, much greater Yb interference (4000 ppm on ^{176}Hf) than the sample material”.

I find that the diagrams are blurry and difficult to read. I think they should explore a bit more the diagrams, pinning down their conclusions more clearly with these diagrams. If there is a problem with manuscript size, I would recommend putting figure 2 in the supplement and redrawing figure 3 so that you have more space for the diagrams.

Our review copy pdf down sampled the figures, in any case we now provide figures as high resolution versions to the journal. Sufficient space exists for us to expand the comparison figures as requested. We have separated out the various Hf patterns from the summary evolution diagram to make our conclusions link more clearly with the figures.

There is too much text about the U-Pb ages, which is hardly used in the discussion. This could also be reduced or put in the supplement.

U-Pb geochronology is discussed in detail in a supplement as it is important to provide the temporal context to this work. Nonetheless, we provide only a short description of the U-Pb data as it is needed to define the evolutionary pathway for the Hf values.

I think the point of this paper is brilliant, but it should be better explained. The authors should focus on what matters, which is the Hf signatures of the NAC and of the Archean rocks around the world. I find it very difficult to see the difference in juvenile addition at 3.2 3.0 and 2.8 Ga in Fig. 4. The symbols are too big and lines are too thick.

We have modified the symbol sizes / line weights and font and believe the figures follow nature guidelines. We also believe this comment may be related to the lower resolution pdf version of the manuscript figures in the review copy.

I have not seen a single spelling mistake, but I presume this should be picked up by the editorial office. I have not seen a single problem with the reference list and with the supplementary data. Finally, I would love to see a more detailed cartoon illustrating how this juvenile crust addition reconcile with models for TTG crust formation. It would be very useful.

We have modified the final summary figure to make it more legible and to more accurately visualise the main points of the paper relating to injection of juvenile material into pre-existing crust during Earth's thermal peak. We do also now provide extra context to TTG production in the text in response to this comment.

REVIEWERS' COMMENTS

Reviewer #1 (Remarks to the Author):

I have re-looked at the manuscript and it now addresses my comments and I am comfortable with how they did this and support publication of the manuscript as it now.

Reviewer #2 (Remarks to the Author):

I have read the MS again and the response letter by the authors. I believe they have made significant changes to accommodate the comments by both reviewers. The paper is in very good shape. I have no further comments to add. Well done!!

Response to reviewers:

Reviewer #1 (Remarks to the Author):

I have re-looked at the manuscript and it now addresses my comments and I am comfortable with how they did this and support publication of the manuscript as it now.

Thank you for taking the time to look at our work and for the constructive comments.

Reviewer #2 (Remarks to the Author):

I have read the MS again and the response letter by the authors. I believe they have made significant changes to accommodate the comments by both reviewers. The paper is in very good shape. I have no further comments to add. Well done!!

Thank you for your contribution to the review process for our work.